# AURO: Reinforcement Learning for Adaptive User Retention Optimization in Recommender Systems

## Abstract

The field of Reinforcement Learning (RL) has garnered increasing attention for its ability of optimizing user retention in recommender systems. A primary obstacle in this optimization process is the environment non-stationarity stemming from the continual and complex evolution of user behavior patterns over time, such as variations in interaction rates and retention propensities. These changes pose significant challenges to existing RL algorithms for recommendations, leading to issues with dynamics and reward distribution shifts. This paper introduces a novel approach called **A**daptive **U**ser **R**etention **O**ptimization (AURO) to address this challenge. To navigate the recommendation policy in non-stationary environments, AURO introduces an state abstraction module in the policy network. The module is trained with a new value-based loss function, aligning its output with the estimated performance of the current policy. As the policy performance of RL is sensitive to environment drifts, the loss function enables the state abstraction to be reflective of environment changes and notify the recommendation policy to adapt accordingly. Additionally, the non-stationarity of the environment introduces the problem of implicit cold start, where the recommendation policy continuously interacts with users displaying novel behavior patterns. AURO encourages exploration guarded by performance-based rejection sampling to maintain a stable recommendation quality in the cost-sensitive online environment. Extensive empirical analysis are conducted in a user retention simulator, the MovieLens dataset, and a live short-video recommendation platform, demonstrating AURO's superior performance against all evaluated baseline algorithms[1].

## Keywords

Reinforcement Learning, recommender systems, user retention, non-stationary environments

## 1 Introduction

Recent advances in recommender systems have shown promising results in enhancing user retention through the application of Reinforcement Learning (RL) [6, 46, 56], primarily due to RL's capacity to effectively manage delayed reward signals [7, 39] and promote efficient exploration strategies [5, 12]. This focus on long-term engagement, as opposed to short-term feedback, is driven by its direct correlation with key performance indicators such as daily active users (DAU) and user dwell time. Nonetheless, the dynamic and ever-changing nature of large-scale online recommendation platforms presents a significant challenge to RL-based recommendation algorithms, with constant shifts in user behavior patterns. For instance, stock traders may exhibit increased activity on a news recommendation application during periods of significant financial

news, and promotions might boost user interactions with recommended products in a shopping application. From the perspective of RL, different user behavior patterns will lead to constantly evolving environment dynamics and reward functions. Due to the poor generalization ability of standard RL methods [25, 47], policies that behave well during training can struggle in the deployment phase due to the evolving recommendation environment.

To improve the generalization ability of RL in dynamic environments, current algorithms in Meta-RL or zero-shot policy generalization [31, 47] attempt to explicitly identify one unique environment parameter (e.g., robot arm masses or joint frictions) as the latent feature corresponding to certain environments. But in the recommendation task, user behaviors exhibit distinct patterns in different time periods, including the positive ratio of immediate feedback, such as click, like, comment, and hate, as well as long-term feedback such as the distribution of user return time, as demonstrated in the motivating example in Sec. 3.1. These factors collectively contribute to a non-stationary recommendation environment, one that cannot be simplified to a single hidden parameter. The non-stationary environment also leads to the challenge of implicit cold start [21], where the recommendation policy continually interacts with users exhibiting new behavior patterns. Therefore, the policy needs to be able to actively adapt to new settings. The exploration ability of some RL methods [5, 12] facilitates such requirement of rapid policy adaptation, but they can be overly optimistic and propose unreliable exploratory actions in the cost-sensitive recommendation environment. To conclude, the dynamic and evolving recommendation environment leads to two critical challenges: to *accurately identify* complex environment changes and to *make safe adaptations* to these changes.

In this work, we propose a new approach termed **A**daptive **U**ser **R**etention **O**ptimization (AURO) that simultaneously addresses both challenges. As RL policies are sensitive to distribution shift [6, 47], the policy performance will be a reliable and universal signal for detecting changes in the recommendation environment. Therefore, to identify non-stationarity actively and universally, we introduce an additional state abstraction module in the policy network, aligning the state abstraction output with state value functions, which serve as proxy performance measure of the current policy. The state abstraction module will then generate different outputs in correspondence with environment changes and notify the learning policy when it is no longer suitable for new user behavior patterns. The alignment between the module's output and policy performance is facilitated by minimizing the discrepancy between the $l_2$-distance in the abstraction space and the estimated performance disparity in the value space. In essence, this loss requires state abstractions to be close to each other on states with analogous values and distant from each other on states with divergent values, providing an unsupervised approach of training the state abstraction network. For rapid policy adaptation, optimism under uncertainty [11, 12] is

---

[1] Our code is available at this anonymous repository.

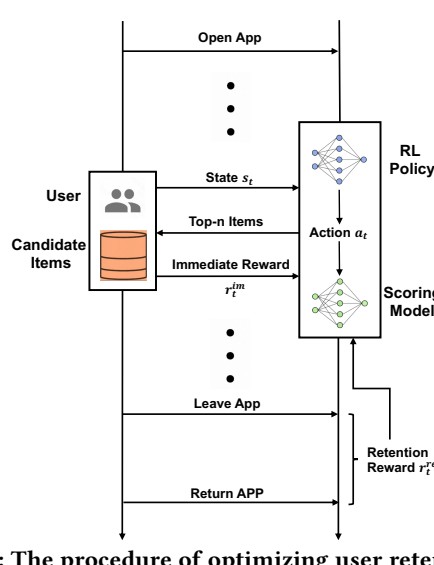

**Figure 1: The procedure of optimizing user retention with RL.**

a common approach for rapid adaptation with exploratory actions. To ensure a safe exploration process, AURO proposes to generate a pool of candidate exploration actions and filter them based on their estimated state-action values. This approach significantly lowers the likelihood of choosing exploration actions that could have adverse effects, thereby enhancing the stability and reliability of the online training process.

To assess the effectiveness of AURO when optimizing user retention in various recommendation tasks, we conduct experiments with a user retention simulator [50] and the MovieLens dataset. We also fully launch AURO in a popular short video recommendation platform. The outcomes of these experiments indicate that AURO outperforms contemporary state-of-the-art methods in both the training stability and the adaptation capacity within complex and non-stationary recommendation environments.

## 2 Background

Fig. 1 illustrates the connection between RL and the practical recommendation process. In this framework, users are treated as environments and the RL policy operates by feeding actions to the environment. As the number of candidate items can be very large and make discrete action selection burdensome, we follow the practice in literature [6, 46] and employ $k$-dimensional continuous actions. At step $t$, the RL policy generates the actions $a_t$. A pre-trained deep scoring model is used to predict a $k$ dimensional score $x_i = (x_{i1}, x_{i2}, \cdots, x_{ik})$ for each candidate item $i$, where each dimension evaluates the item in a particular aspect. The scoring model will be treated as a black box in this paper. Subsequently, a pre-defined ranking function $f$ is employed to compute the final ranking score $f(a_t, x_i)$ for each selected item $i$. The system then recommends the top-$n$ items to the user according to the ranking score.

According to the formulation in Fig. 1, fluctuations in user behavior patterns can make the recommendation environment non-stationary. To model the recommendation task in such environments, we employ the Hidden Parameter Markov Decision Process

**Table 1: Statistics of the dataset sampled in the live environment. The immediate feedbacks are normalized to have unit mean before computing the standard deviation.**

| Dimension | Number | Positive Ratio | Normalized Standard Deviation of Ratio |
|---|---|---|---|
| Users | 27,285 | - | - |
| Items | 7,551 | - | - |
| Samples | 1,186,059 | - | - |
| Click | 208,934 | 17.61% | 0.049 |
| Like | 5,691 | 0.480% | 0.116 |
| Follow | 310 | 0.026% | 0.286 |
| Comment | 411 | 0.035% | 0.418 |
| Hate | 1,351 | 0.114% | 0.334 |

(HiP-MDP) [14] $< \mathcal{S}, \mathcal{A}, \Theta, T, r, \gamma, \rho_0 >$ that attributes the non-stationarity to an agnostic hidden parameter distribution $\Theta$. In the HiP-MDP, $\mathcal{S}$ is the continuous state space. The state $s_t \in \mathcal{S}$ contains user profile, user interaction history, and candidate item information at timestep $t$. $\mathcal{A}$ is the $k$-dimensional continuous action space. $\Theta$ is the joint hidden parameter distribution with $n$ variables, where $n$ is the number of unique hidden parameters. When the episode starts, $\theta \sim \Theta$ is sampled reflecting the current status of the non-stationary environment. Both $\Theta$ and $\theta$ are invisible to RL algorithms. $T$ is the transition function. At step $t$, $T$ updates the user profile and browsing history in state $s_t$ according to the action $a_t$ and current hidden parameters $\theta$, generating the next state distribution $s_{t+1} \sim T_\theta(\cdot|s_t, a_t, \theta)$. $r$ is the reward function. To optimize user retention, the time gap of users returning to the recommendation platform is used to calculate the retention reward assigned to the last step of an episode $r_\theta(s_0, a_0, s_1, a_1, \cdots, s_t, a_t, \theta) = \lambda \times$ user return time, where $\lambda$ is the predefined weighing coefficient. The reward is set to zero at other steps. $\gamma$ is the discount factor that determines how much the policy accounts for rewards in future steps. $\rho_0$ is the initial state distribution. When the session starts, $s_0$ is randomly sampled according to $\rho_0$ and the current hidden parameter $\theta$: $s_0 \sim \rho_0(\cdot|\theta)$. We aim at maximizing the expected accumulated return of the policy $\pi$: $\eta_T(\pi) = E_{\pi,T,\Theta}[\sum_{t=0}^{\infty} \gamma^t r_\theta(s_0, a_0, \cdots, s_t, a_t, \theta)]$, where the expectation is computed with $a_t \sim \pi(\cdot|s_t)$, $s_{t+1} \sim T_\theta(\cdot|s_t, a_t, \theta)$, and $\theta \sim \Theta$. The state-action value function $Q^\pi(s, a)$ denotes the expected return after taking action $a$ at state $s$: $Q^\pi(s, a) = E_{\pi,T,\Theta} \sum_{t=0}^{\infty} \gamma^t r_\theta(s, a, \cdots, s_t, a_t, \theta)$, also referred to as the Q-function. The state value function, or the V-function, is defined as $V^\pi(s) = \mathbb{E}_{a \sim \pi(\cdot|s)} Q^\pi(s, a)$.

## 3 AURO: Adaptive User Retention Optimization

### 3.1 Motivating Example

The main focus of this paper is to address the challenge of non-stationary environment arisen from multiple fluctuation factors of user behaviors. To empirically justify the existence of such issue in practical recommendation systems, we conduct a data-driven study in a popular short-video recommendation platform, which is also used in live experiments in Sec. 4.3. To avoid the influence of different recommended items and interaction histories, we select interactions that occur at the beginning of each recommendation session, i.e., the $s_0$ of the trajectory. For the purposes of this paper, we are primarily interested in investigating the users' return-time

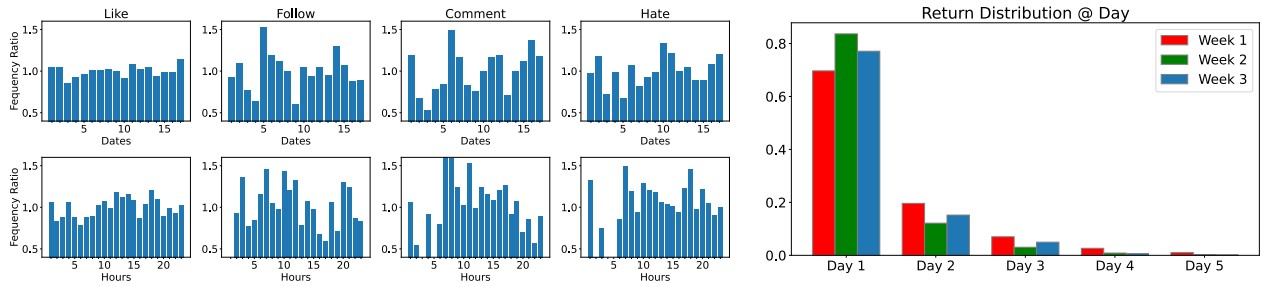

**Figure 2: Left: Normalized ratio of immediate user feedback among all data samples in the live environment, compared on different dates in one month and different hours in one day. Right: The distribution of user return time in three consecutive weeks. The return probabilities at different days exhibit variability over time.**

distribution and the positive ratio of immediate feedback among all data, including click, like, follow, comment, and hate.

We visualize the normalized interaction ratio across different dates and hours in Fig. 2 (left). The distribution of the user return time in three weeks is also illustrated in Fig. 2 (right). We also demonstrate the normalized standard deviations of several immediate interaction ratio in Tab. 1. The return probability and interaction ratio exhibit large variability over time. For example, the average probability of user returning to the application in the next day can be as low as about 70% in week 1, and as high as about 81% in week 2. Among immediate signals, the "click" and "like" signals exhibit relatively more stability, while the other three signals fluctuates significantly, with 3 to 4 times higher standard deviation than "like". For example, the "follow" signal is about twice more frequent on day 5 than on day 9 and the "comment" signal is three times more frequent on 8 a.m. than on 3 and 5 a.m. Furthermore, we can hardly identify any clear pattern in the changes of interaction frequency between dates within a month or hours within a day. Therefore, it becomes challenging to manually extract environment information with rules and feed to the policy network. Overall, these findings characterize the complex and non-stationary nature of the recommendation environment, where several different aspects of user behaviors evolve over time.

## 3.2 Framework Overview

To tackle the aforementioned challenges, we introduce a new paradigm called **A**daptive **U**ser **R**etention **O**ptimization (AURO). Its architecture is shown in Fig. 3 and the algorithm procedure is listed in Appendix C. As discussed in Sec. 2, the recommendation policy takes user features, item features, and item history as input. They are processed by embedding networks and transformers, before being concatenated together and generating the vectorized state. In the following part of this section, we discuss two key contributions of AURO, namely the state abstraction network and the guarded online exploration procedure.

## 3.3 Universal Non-stationarity Identification with Value-based State Abstraction

In this paper, we introduce an extra state abstraction network and align its outputs with the expected performance of the current policy, which is more universal in reflecting environment non-stationarity. When a new hidden parameter $\theta$ in HiP-MDP is sampled, the policy performance will be degraded due to dynamics

and reward distribution shift, and therefore change accordingly with the evolvement of the dynamic environment. The output of a well-aligned state abstraction network can then serve as a universal approach of identifying the current environment status in face of non-stationarity. To achieve the abstraction-performance alignment, we enforce the $l_2$-distance in the state abstraction space to be close to a performance-related distance measure $d$ that reflects policy performance. The loss function for updating the state abstraction network $\phi$ can then be expressed as

$$J(\phi) = \sum_{i,j} \left[ \left\| \phi\left(s_i\right) - \phi\left(s_j\right) \right\|_2 - d\left(s_i, s_j\right) \right]^2. \tag{1}$$

In the Actor-Critic architecture of RL, the state value function $V$ can serve as the critic and can evaluate the performance of the learning policy. If both two states $s_i, s_j$ have high state values, the policy will perform well on both of them, so the latent variable $\phi(s_i)$ should be close to $\phi(s_j)$. If two states have different state values, their corresponding latent variables should be far from each other. Therefore, we choose the following distance measure based on the state value function $V$:

$$d\left(s_i, s_j\right) = \begin{cases} 0 & \text{if } V\left(s_i, \phi(s_i)\right) \text{ and } V\left(s_j, \phi(s_j)\right) \text{ are close,} \\ \infty & \text{otherwise.} \end{cases} \tag{2}$$

By setting the distance function to have extreme values with different inputs of estimated policy performances, the state abstractor $\phi(s)$ will be provided with more precise and simplified signals to discriminate different environments. As illustrated in Fig. 3, the state abstraction $\phi(s)$ also serves as an additional input to the state-value function $V$ for more accurate value estimation. The comparison of value error[2] during training between different algorithms is in Fig. 6 (d). The value estimation in AURO is more accurate than other algorithms without state abstraction. This justifies the feasibility of using state value functions as proxies of policy performance. Meanwhile, lower value error corresponds to better training stability in face of non-stationary environments, demonstrating the effectiveness of the additional state abstraction module.

To determine whether $V(s_i, \phi(s_i))$ and $V(s_j, \phi(s_j))$ are close, we rank a batch of input states $s_1, s_2, \cdots, s_B$ with size $B$ by their state values and divide them into $n$ categories $C_1, C_2, \cdots, C_n$, where $n$ is a hyperparameter. States that are assigned to the same category

---

[2]The value error $\mathcal{E}_t$ is computed according to the Bellman error: $\mathcal{E}_t = [r(s_t, a_t, \theta) + \gamma V(s_t) - V(s_{t+1})]^2$ on 10,000 randomly selected states in the replay buffer.

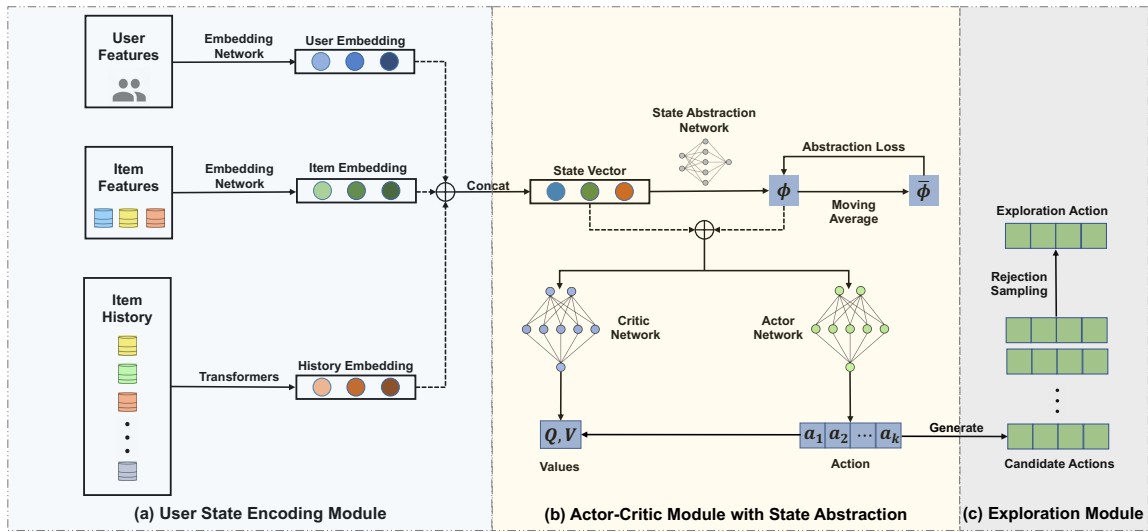

**Figure 3: Overview of the AURO framework. (a) The user state encoding module that embed user features, item features, and the item history into a low-dimentional state vector. (b) The actor-critic module with a state abstraction network that generates the latent feature vector $\phi(s)$. The state vector is concatenated with $\phi(s)$ before serving as the input to the actor and critic networks. (c) The exploration module for selecting exploration actions that interact with the recommendation environment.**

are considered to have similar values. We denote $j \in N(i)$ if $s_i$ and $s_j$ fall into the same category. Plugging Eq. (2) into the original loss function Eq. (1), we get[3]

$$J(\phi) = \sum_i \left[ \sum_{j \in N(i)} \left\| \phi(s_i) - \phi(s_j) \right\|_2^2 - \sum_{j \notin N(i)} \left\| \phi(s_i) - \phi(s_j) \right\|_2^2 \right], \tag{3}$$

where the first term makes state abstractions in the same category closer, and the second term pushes them in different categories away from each other.

In practice, the state-value function $V$ is updated alongside policy training and the state batch is randomly sampled from the replay buffer that keeps updating. Therefore, the output $\phi(s_i)$ can be unstable, which is undesirable when using $\phi(s_i)$ as part of the policy input. To mitigate this problem, we incorporate the moving average $\tilde{\phi}_k = (1 - \eta)\tilde{\phi}_k + \frac{\eta}{|C_k|} \sum_{s_i \in C_k} \phi(s_i), k = 1, 2, \cdots, n$ of each state categories to both terms in Eq. (3). We transform Eq. (3) to make it related with the average state abstraction $\bar{\phi}_k = \frac{1}{|C_k|} \sum_{s_i \in C_k} \phi(s_i)$. With regard to the first term (denoted as $J_{\text{same}}(\phi)$), we have

$$J_{\text{same}}(\phi) = \sum_{k=0}^n \sum_{s_i, s_j} \left\| \phi(s_i) - \phi(s_j) \right\|_2^2 = \frac{2B}{n} \sum_{k=0}^n \sum_{s_i} \left\| \phi(s_i) - \bar{\phi}_k \right\|_2^2.$$

To maximize $J(\phi)$ with states from different categories (denoted as $J_{\text{diff}}(\phi)$), we have

$$J_{\text{diff}}(\phi) = -\sum_{k=0}^n \sum_{m>k} \sum_{s_i \in C_k} \sum_{s_j \in C_m} \left\| \phi(s_i) - \phi(s_j) \right\|_2^2$$

$$\leqslant -\frac{B^2}{n^2} \sum_{k=0}^n \sum_{m>k} \left\| \bar{\phi}_k - \bar{\phi}_m \right\|_2^2. \tag{4}$$

---
[3]Detailed derivations of equations in this section are listed in Appendix A.

By minimizing the last term in Eq. (4), we are minimizing an upper-bound of the original loss function. The average state abstraction $\bar{\phi}_k$ can be replaced with the moving average $\tilde{\phi}_k$, giving rise to the final loss function that is used during training:

$$J(\phi) = \frac{2B}{n} \sum_{k=0}^n \sum_i \left\| \phi(s_i) - \tilde{\phi}_k \right\|_2^2 - \frac{B^2}{n^2} \sum_{k=0}^n \sum_{m>k} \left\| \tilde{\phi}_k - \tilde{\phi}_m \right\|_2^2. \tag{5}$$

In practice, in addition to the value-based loss function $J(\phi)$, the policy optimization loss is also backpropagated to the state abstraction network during training to accelerate the training process.

## 3.4 Guarded Online Exploration for Implicit Cold Start

In the non-stationary recommendation environment illustrated in Sec. 3.1, the recommendation policy is continuously facing users with new behavior patterns. From the perspective of HiP-MDP, different $\theta$ at different episodes will lead to new environment dynamics $T$ and reward functions $r$. Although the state abstraction network in Sec. 3.3 can help policies *identify* environment novelty, the policy itself needs to be able to actively *adapt* to new environment parameters. This is similar to the well-known cold-start issue in recommendation [23, 34], but happens implicitly during online training.

An ideal approach to deal with such issue in RL is exploration. We consider the approach of optimism under uncertainty [12] with the optimistic state-action value estimation, defined as $Q_{\text{UB}}(s, a) = \mu_Q(s, a) + \beta \sigma_Q(s, a)$, where $\mu_Q$ and $\sigma_Q$ represent the mean and standard deviation of the outputs from the Q networks, respectively, and $\beta$ is a hyper-parameter. The exploration action $a_E$ can be calculated by extending the original policy output $a_T$ in the direction

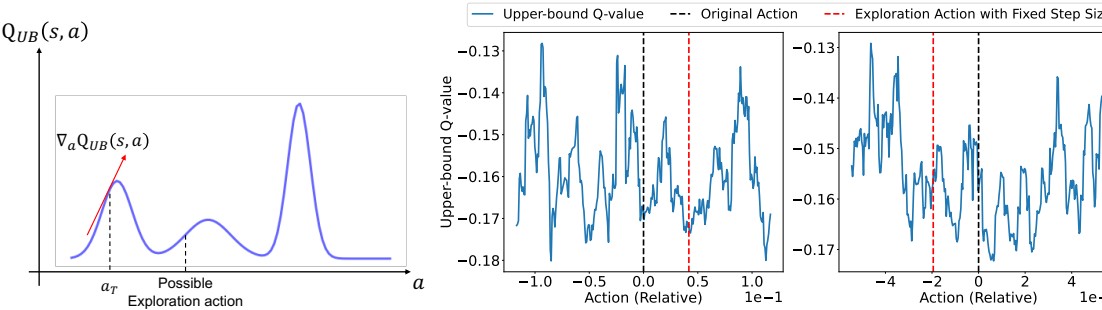

**Figure 4: Left: The demonstration of action selection with optimism under uncertainty in Eq. 6. The action can miss the local optimums of the state-action value function; Right: The state-action value function on two example state-action pairs in the user retention simulator [50]. The vertical lines show the relative values of the original and exploration action in one dimension. The exploration action generated by Eq. 6 with a fixed step size $\delta$ can lead to a lower value than the original action.**

of gradient ascent of $Q_{\text{UB}}(s, a)$ [12]:

$$a_E = a_T + \delta \cdot \frac{[\nabla_a Q_{\text{UB}}(s, a)]_{a=a_T}}{\| [\nabla_a Q_{\text{UB}}(s, a)]_{a=a_T} \|}, \quad (6)$$

where $[\nabla_a Q_{\text{UB}}(s, a)]_{a=a_T}$ is the gradient of $Q_{\text{UB}}$ with respect to the action $a_T$, and $\delta$ is the step size hyperparameter. However, determining the extent to which the original action should be extended, i.e., the step size $\delta$, can be challenging, as a small step size may lead to inefficient exploration, and a large step size can result in inaccurate gradient approximation. As illustrated in Fig.4 (left), the state-action function can exhibit multiple peaks, and an improper step size may cause the exploration action to miss a local optimum.

The challenge of selecting an appropriate step size $\delta$ is more pronounced in recommendation tasks, as the landscape of state-action values in such tasks can exhibit high complexity, which is illustrated in Fig. 4 (middle and right). Another challenge is that recommendation tasks can be risk-sensitive: users may disengage from the application and cease the recommendation process if they encounter recommended items that fail to capture their interest. To mitigate the aforementioned issues, we propose to generate a set of actions with optimistic exploration and filter out candidate actions that may lead to poor recommendation quality with rejection sampling. The actions $a_k, k = 1, 2, \cdots, N$ are located near the original policy output $\pi(s, \phi(s))$, in the direction of the gradient $\nabla_a Q_{\text{UB}}(s, a)$. The exploration action $a_E$ is selected form the action set according to the average Q-value:

$$a_E = \arg\max_{a_k} \mu_Q(s, a_k), \quad a_k = a_T + k\delta [\nabla_a Q_{\text{UB}}(s, a)]_{a=\pi(s)} \quad (7)$$

where $\delta$ is the hyper-parameter controlling the gap of action particles. It can be set to a small value and does not need extra tuning, as shown by the ablation study in Tab. 2. By choosing from several candidate actions, the exploration module manages to find actions with higher state-action values more efficiently and reduces the risk of adopting dangerous actions that have low values. We also visualize the effectiveness of this exploration technique in Fig. 6.

## 4 Experiments

To evaluate and analyse the practical performance of AURO, we conduct extensive experiments to investigate the following research

questions (RQs): RQ1: Can AURO optimize user retention better than baseline algorithms, when applied to recommendation datasets and simulators? RQ2: How does each component of AURO contribute to overall performance? RQ3: Can AURO perform well in online A/B tests of large-scale live recommendation platforms? To answer these questions, the state abstraction module in AURO is used to generate recommendation policies in the modified MovieLens-1M dataset and the KuaiSim retention simulator [50]. We conduct ablation studies and visualizations to investigate the contribution of each component of AURO to the overall performance. AURO is also deployed in a dynamic, large-scale, real-world short video recommendation platform to perform live A/B test.

### 4.1 Experiments in Retention Simulator

**Setup** AURO adopts a novel paradigm that focuses on optimizing user retention, rather than immediate user feedback. However, recommendation simulators that have been widely used [20, 36, 42] cannot simulate user retention behaviors. The KuaiSim retention simulator [50] aims to simulate long-term user retention on short video recommendation platforms. It has verified several recommendation algorithms [6, 29, 30] that also investigate user retentions. The KuaiSim simulator contains a user leave module, which predicts whether the user will leave the session and terminate the episode; and a user return module, which predicts the probability of the user returning to the platform on each day as a multinomial distribution. The probabilities of user leaving and returning are altered in each episode to capture the dynamic nature of user behaviors and create a non-stationary evaluation environment. More information on the simulator setup is in Appendix D.1.

For baseline algorithms, we include non-RL recommendation methods (CEM [13], DIN [53]), state-of-the-art value-based RL algorithms (DDPG [26], TD3 [18], SAC [19]), RL algorithms facilitating efficient exploration (OAC [12], RND [5]), context encoder-based RL algorithms (ESCP [31]), and RL-based recommendation algorithms for optimizing user retention (RLUR [6], OSPIA [44], RL-LTV [21]). Detailed descriptions of the baseline algorithms are in Appendix D.2. In simulator-based experiments, we choose three metrics to evaluate the algorithms: the users' average return days (lower is better), the users' return probability on the next day (higher is better), and

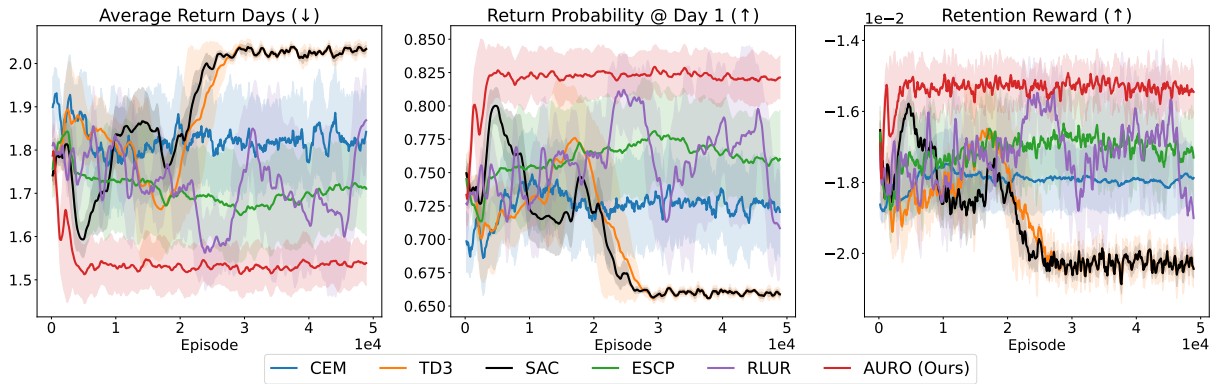

**Figure 5: Performance comparison of different algorithms in the modified KuaiSim simulator. Metrics with the up arrow (↑) are expected to have larger values and vice versa.**

**Table 2: Left: Performance comparisons of AURO and all the baseline algorithms in the simulator. The scores are computed at timestep 20K. Right: Results of ablation studies. $n$ is the number of categories in state abstraction loss Eq. (5); $\delta$ is the action partition gap when computing the exploration action according to Eq. (7).**

| | Average Return Days (↓) | Return Rate at Day 1 (↑) | Retention Reward (↑) |
|---|---|---|---|
| CEM | 1.858±0.232 | 0.715±0.065 | -0.018±0.001 |
| DIN | 1.725±0.029 | 0.755±0.005 | -0.017±0.001 |
| TD3 | 1.772±0.146 | 0.738±0.048 | -0.017±0.001 |
| SAC | 1.810±0.166 | 0.726±0.053 | -0.017±0.001 |
| DDPG | 2.012±0.013 | 0.662±0.003 | -0.019±0.000 |
| OAC | 1.794±0.070 | 0.731±0.026 | -0.018±0.000 |
| RND | 1.828±0.034 | 0.724±0.012 | -0.018±0.000 |
| ESCP | 1.705±0.125 | 0.764±0.039 | -0.017±0.001 |
| RL-LTV | 1.719±0.281 | 0.761±0.050 | -0.017±0.003 |
| RLUR | 1.706±0.114 | 0.765±0.038 | -0.017±0.001 |
| OSPIA | 1.726±0.113 | 0.803±0.020 | -0.017±0.001 |
| AURO (Ours) | **1.531**±0.058 | **0.824**±0.018 | **-0.015**±0.000 |

| | Average Return Days (↓) | Return Rate at Day 1 (↑) | Retention Reward (↑) |
|---|---|---|---|
| No exploration | 1.868±0.061 | 0.708±0.015 | -0.018±0.000 |
| No state abstraction | 1.803±0.109 | 0.732±0.034 | -0.018±0.001 |
| No auxiliary loss | 1.672±0.208 | 0.776±0.068 | -0.017±0.002 |
| $n = 3$ | 1.598±0.171 | 0.804±0.029 | -0.016±0.002 |
| $n = 5$ | 1.534±0.062 | 0.815±0.011 | -0.015±0.001 |
| $n = 6$ | 1.550±0.064 | 0.812±0.011 | -0.016±0.001 |
| $\delta = 0.3$ | 1.547±0.068 | 0.813±0.013 | -0.016±0.001 |
| $\delta = 3$ | 1.565±0.125 | 0.808±0.023 | -0.016±0.001 |
| AURO ($n = 4, \delta = 1$) | 1.531±0.058 | 0.824±0.018 | -0.015±0.000 |

the retention reward defined in Sec. 2 (higher is better). All algorithms for comparison are run for 50,000 training steps with five different random seeds.

**Performance Comparison** The training curve for AURO, as well as baseline algorithms CEM, TD3, SAC, ESCP, and RLUR, are shown in Fig. 5. The table for comparisons with all baseline algorithms at timestep 20K is in Tab. 2 (left). CEM and DIN perform worse than the other RL-based algorithms. This is because supervised-learning algorithms can only effectively learn from immediate response and highlights the effectiveness of RL in optimizing user retention. The performance of TD3 and SAC exhibits improvements in the early stage of training, but deteriorates as training proceeds. Without explicit modeling of the environment distribution shift, the policies they obtain are loosely coupled with specific user behavior patterns, leading to suboptimal performance. The exploration algorithms OAC and RND do not exhibit large performance improvements compared to TD3 and SAC, largely due to their unsafe exploration procedure. RLUR and OSPIA take into account the bias of long-term recommendation optimization and outperform TD3 and SAC. But they suffer from unstable training and take more steps to converge than AURO. The ESCP algorithm

incorporates a context encoder that can capture the environment non-stationary to some extent. But it explicitly relies on a single hidden parameter, and cannot model environment fluctuations universally. As a result, it has a stable training curve, but exhibits suboptimal overall performance.

Compared with baseline algorithms, AURO shows a stable training curve and the best overall performance. The stability is due to the state abstraction module which enables the algorithm to fit different environment dynamics and reward functions. The good asymptotic performance can be attributed to the safe and efficient exploration module that quickly navigates to high-reward regions when the environment changes.

**Ablations and Analyses** We conduct ablation studies in the modified KuaiSim simulator to analyze the role of each module in AURO, namely the state abstraction network, the value-based contrastive loss to train the network, and the selection of exploratory actions. We also investigate the effect of different values of hyperparameters $n$ in the value-based loss function and $\delta$ in selecting exploratory actions. As shown in Tab. 2 (right), removing any of these components will lead to a drop in the overall algorithm's performance. Among them, the exploration module has the most

Figure 6: (a): Visualizations of the state abstraction output in Sec. 3.3 with dimension reduction. States with different values are assigned with different colors. (b) and (c): The state-action value function on two example state-action pairs. The vertical lines show the relative values of the original action, the exploration action with a fixed step size, and the AURO exploration action. (d): Comparison of value loss between AURO and baseline algorithms. AURO has a more accurate value network thanks to the state abstraction module.

Table 3: Performance comparison of different algorithms in the modified MovieLens-1M dataset.

|  | BC | CEM | TD3+BC | ESCP+BC | RLUR+BC | AURO (Ours)+BC |
|---|---|---|---|---|---|---|
| Retention Reward | $1.702 \pm 0.018$ | $1.708 \pm 0.055$ | $1.714 \pm 0.054$ | $1.755 \pm 0.031$ | $1.784 \pm 0.060$ | $\mathbf{1.798 \pm 0.032}$ |

significant impact. The comparison emphasizes the necessity of exploration when optimizing the sparse retention signal. We also analyse the computation cost of AURO's exploration module in Appendix D.3. The performance of AURO will also decrease without state abstraction or training the state abstraction network only with the policy loss (without the value-based auxiliary loss). This demonstrates the effectiveness of the state abstraction module and the new value-based loss function proposed in Sec. 3.3. Different values of $n$ or $\delta$ will make little influence on AURO's performance, demonstrating its robustness to hyperparameter selections.

We visually demonstrate two of the key components of the AURO algorithm in Fig. 6. Fig. 6 (a) illustrates the outputs of the state abstraction module with a batch of states as input. The outputs are projected into two dimensions with PCA for visualization and colored according to the state values. As shown in the figure, states with similar values exhibit closely gathered abstractions , while those in different value categories tend to have distinct abstractions. In this way, the state abstraction can help the learning policy identify whether it will perform well in the current environment. Fig. 6 (b,c) show the landscapes of state-action values when the action is altered in one dimension. Our exploration policy can find a better action (vertical green line) than the exploration action generated with a fixed step size (vertical red line). Fig. 6 (d) shows the comparison of value loss baetween AURO and baseline algorithms, where AURO has the lowest value error during training. This demonstrates the effectiveness of the state abstraction module in enhancing value estimation.

## 4.2 Experiments in MovieLens Dataset

To verify the effectiveness of AURO in different recommendation tasks, we make evaluations with the well-known recommendation dataset MovieLens-1M. It contains 1,000,209 anonymous ratings of approximately 3,900 movies, generated by 6,040 MovieLens users. We sample the data of 5000 users as the training set, and use the data of the remaining users as the test set.

Following previous researches in user retention optimization [43, 46], we assume that average user return days is proportional to the movie ratings in MovieLens-1M. We assign retention reward of -1 with rating score 1, and retention rewards 1,2,3 for rating scores 3,4,5, respectively. To ensure conservative policy update during offline training, we remove the exploration module and add BC loss when training policies with online RL algorithms, similar to the TD3+BC [16] algorithm. We use Normalised Capped Importance Sampling (NCIS) [41], a standard offline evaluation method, to make offline algorithm evaluations. The comparative results are listed in Tab. 3, where AURO achieves the highest performance. This demonstrates the effectiveness of the state abstraction module in state representation even without explicit changes in user behaviors.

## 4.3 Experiments in Live Experiments

**Setup** The MDP setup in the live experiments is similar to that described in Sec. 2. The algorithms are incorporated in a candidate-ranking system of a popular short-video recommendation platform. The live experiment is run continuously for two weeks. It involves an average of 25 million active users and billions of interactions each day. With such long time period and large scale of involved users, the recommendation environment can exhibit large deviations, as shown by the analysis in Sec. 3.1 based on data collected in this platform. For comparative analysis, users are randomly split into several buckets. The first bucket runs the baseline algorithm RLUR, and the remaining buckets run algorithms AURO, TD3, and ESCP. We do not consider as many baseline algorithms as in dataset or simulator based experiments, because online experiments are cost-sensitive and poor algorithms can potentially bore or annoy platform users. In live experiments, the retention signal can be noisy and sometimes influenced by other module of the live platform. Therefore, the main metric we use is the rate of users returning to the platform averaged between 7 days, rather than the 1-day retention rate. We also focus on the application dwell time, as well as immediate user responses including video click-through rate (CTR)

**Table 4: Performance comparison of different algorithms with the RLUR baseline in live experiments. Metrics with the up arrow (↑) are expected to have larger values and vice versa.**

|  | 7d Retention Rate ‰, ↑ | Dwell Time ‰, ↑ | Click-through Rate ‰, ↑ | Like Rate ‰, ↑ | Comment Rate ‰, ↑ | Unlike Rate %, ↓ |
|---|---|---|---|---|---|---|
| TD3 | 0.041±0.148 | -0.685±0.297 | 1.412±0.619 | 1.798±1.127 | 1.715±1.782 | -0.567±0.820 |
| ESCP | 0.123±0.073 | -0.115±0.277 | 0.703±0.313 | 0.302±0.952 | -1.975±1.204 | -0.116±0.791 |
| AURO (Ours) | **0.138**±0.089 | **0.263**±0.181 | **3.260**±0.332 | **2.821**±0.925 | **8.392**±1.881 | **-1.874**±0.781 |

(click and watch the video), like rate (like the video), comment rate (provide comments on the video), and unlike rate (unlike the video). These metrics are standard evaluation criteria for recommendation algorithms and are empirically shown to be related to long-term user experiences [43].

**Performance Analysis**    The comparative results are shown in Tab. 4. The statistics are permillage or percentage improvements compared with RLUR. AURO exhibits superior performance in all evaluation metrics than baseline algorithms, including TD3, ESCP, and RLUR. Specifically, AURO is the only algorithm that achieves performance improvement in the application dwell time. AURO also improves the rate of user comments by 8.392‰, which is almost 5 times larger than the improvements of TD3. These empirical results demonstrate AURO's effectiveness and scalability when applied to live recommendation platforms.

From a practical perspective, some naive algorithm modifications in live environments can hardly simultaneously improve all performance metrics in Tab. 4. This phenomenon has been reported in literature [7] and is also observed in our live experiment. In Tab. 4 the ESCP algorithm can increase CTR and like rate, but witness drops in dwell time and comment rate. In contrast, our AURO algorithm is capable of improving all performance metrics, meanwhile highlighting the large drop in hate rate. This can be because previous approaches only recommends common items that lead to high CTR/like rate, such as short videos that are very amusing or goods that are very cheap. Meanwhile they have limited abilities in recommending user-oriented items that really consider the dynamic behavior of each user, as demonstrated by the poor user dwell time and comment rate. Thanks to our state abstraction module for policy adaptation, our AURO algorithm can recommend up-to-date, user-oriented items that captures the dynamic user interests. The large drop in AURO's user hate rate provides evidence for such inference.

## 5   Related Work

**RL for Recommendation**    In Reinforcement Learning (RL) [40], a learning agent interacts with the environment [32, 37] or exploits offline dataset [17] to optimize the cumulative reward obtained throughout a trajectory. RL has been found promising for recommendation tasks [1, 9, 35, 49]. Traditional approaches propose to learn an additional world model [2] or explicitly construct high-fidelity simulators [36] to enhance the sample efficiency of RL algorithms. They can also be categorized into value-based [9, 27] or policy-based [8] methods. Recently, increasing attention has been paid to optimizing user retention with RL [56]. RLUR [6] and OSPIA [44] deal with issues that comes along in this field, including delayed reward, uncertainty, and instability of training. They will

also serve as baselines in the experiments. Other approaches focus on reward engineering, employing the constrained actor-critic method [7] or incorporating human preferences [45]. Our paper also aims at optimizing user retention and focuses on mitigating the challenge of non-stationary environments that has largely been ignored in previous methods.

**Adaptive Policy Training with RL**    There are a handful of RL algorithms that train adaptive policies in evolving environments with distribution shift. Meta-RL algorithms [3, 33] can adapt to new environments by fine-tuning a small amount of data from the test environment. Zero-shot adaptation algorithms like ESCP [31], CaDM [24] and SRPO [47] can fit new environments without additional data. A key component enabling rapid adaptation of RL policies is the context encoder [31], which takes a stack of history states as input and generates a dense vector that represents different contexts. The encoder can be trained with variational inference [15, 55] or auxiliary losses [31]. Its output can be used as additional inputs to the environment model [24] or the policy network [10]. Other approaches focus on representation learning [48] or importance sampling [28]. However, none of these methods specifically consider the complexity of the recommender systems and can be unreliable or inefficient when applied directly.

We leave relevant researches on recommendation with evolving user interests in Appendix B.

## 6   Conclusion

In this work, we address the challenge of non-stationary environments when optimizing user retention with RL. Through data-driven analyses, we identify several fluctuation factors in user behavior, such as the positive ratio of immediate feedback and the user return time distribution, as the main causes of non-stationarity. To tackle this challenge, we propose the AURO algorithm for training adaptive recommendation policies. AURO utilizes a new value-based contrastive loss to train an additional state abstraction module in the policy network. The state abstraction enables RL policies to identify different sorts of fluctuations in user behavior patterns universally. To facilitate safe and efficient policy adaptation, we combine the idea of optimism under uncertainty with rejection sampling to boost exploration. Experimental results demonstrate that AURO outperforms state-of-the-art methods in optimising long-term and non-stationary user retention. It also ensures stable recommendation quality in face of distribution shift of environment dynamics and reward function.

**Ethics Statement**    While our approach involves real user logs, the user data we used have been stripped of all sensitive privacy information. Each user is denoted by an anonymous user id, and sensitive features are encrypted in the form of one-hot vectors.

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

## A  Detailed Derivations

The loss $J_{\text{same}}(\phi)$ can be derived as follows:

$$
\begin{aligned}
J_{\text{same}}(\phi) &= \sum_{k=0}^{n} \sum_{s_i,s_j \in C_k} \left\| \phi(s_i) - \phi(s_j) \right\|_2^2 \\
&= \sum_{k=0}^{n} \left[ \sum_{s_i \in C_k} 2|C_k| \|\phi(s_i)\|_2^2 - 2 \sum_{s_i,s_j \in C_k} \phi^T(s_i)\phi(s_j) \right] \\
&= \frac{2B}{n} \sum_{k=0}^{n} \sum_{s_i \in C_k} \phi^T(s_i)(\phi(s_i) - \bar{\phi}_k) \\
&= \frac{2B}{n} \sum_{k=0}^{n} \sum_{s_i \in C_k} \phi^T(s_i)(\phi(s_i) - \bar{\phi}_k) - \bar{\phi}_k^T(\phi(s_i) - \bar{\phi}_k) \\
&= \frac{2B}{n} \sum_{k=0}^{n} \sum_{s_i \in C_k} \left\| \phi(s_i) - \bar{\phi}_k \right\|_2^2 .
\end{aligned}
\tag{8}
$$

The loss $J_{\text{diff}}(\phi)$ can be derived as follows:

$$
\begin{aligned}
J_{\text{diff}}(\phi) &= -\sum_{k=0}^{n} \sum_{m>k} \sum_{s_i \in C_k} \sum_{s_j \in C_m} \left\| \phi(s_i) - \phi(s_j) \right\|_2^2 \\
&= -\sum_{k=0}^{n} \sum_{m>k} \sum_{s_i \in C_k} \left[ \frac{1}{|C_m|} \sum_{s_j \in C_m} \left\| \phi(s_i) - \phi(s_j) \right\|_2^2 \sum_{s_j \in C_m} 1^2 \right] \\
&\leqslant -\sum_{k=0}^{n} \sum_{m>k} \sum_{s_i \in C_k} \left[ \frac{n}{B} \left\| \sum_{s_j \in C_m} \left[ \phi(s_i) - \phi(s_j) \right] \right\|_2^2 \right] \\
&\leqslant -\frac{B}{n} \sum_{k=0}^{n} \sum_{m>k} \sum_{s_i \in C_k} \left\| \phi(s_i) - \bar{\phi}_m \right\|_2^2 \\
&\leqslant -\frac{B^2}{n^2} \sum_{k=0}^{n} \sum_{m>k} \left\| \bar{\phi}_k - \bar{\phi}_m \right\|_2^2 .
\end{aligned}
\tag{9}
$$

## B  Additional Related Work

### B.1  Recommendation with Evolving User Interests

Previous point-wise or list-wise recommendation models have incorporated the concept of evolving user behavior, primarily focusing on changes in the distribution of user interests [22, 38]. However, our paper considers other user behaviors related to the long-term user experience, such as the rate of immediate response and the distribution of user return time. To model the evolving user interests, [53, 54] learn the representation of user interests from historical behaviors and performs click-through rate prediction. [4] proposes to ensure that sufficiently diversified content is recommended to the user in face of adaptive preferences. But the algorithm is analysed in the bandit setting without empirical justifications. [52] predicts users' shift in tastes during training and incorporates these predictions into a post-ranking network. Another approach involves predicting the distributions of the top-k target customers and training the recommendation model accordingly [51]. It is also worth noting that some studies have indicated that these forms of distribution adjustment may negatively impact the overall recommendation accuracy [51]. Instead of predicting user behaviors and train recommendation models beforehand, our paper focus on the identification of new distributions and the ability to rapidly adapt to them.

---

**Algorithm 1** The workflow of AURO

1: **Input:** The state abstraction network $\phi_\varphi$, the deterministic policy $\pi_\theta$, the state-action value function $Q_\psi$, the replay buffer $D$, training steps $N$, and the training horizon $H$.
2: Initialize the networks and the replay buffer.
3: **for** $1, 2, 3, \ldots, N$ **do**
4:     **for** $t = 1, 2, \ldots, H$ **do**
5:         Obtain $z_t$ from $\phi_\varphi(s_t)$ and sample $a_T$ from $\pi_\theta(s_t, z_t)$.
6:         Get the exploration action $a_E$ with Eq. (7) and set $a_t = a_E$.
7:         Interact with the simulator, get transition data $(s_{t+1}, r_t, d_{t+1}, s_t, a_t, z_t)$, and add it to $D$.
8:     **end for**
9:     Update the state abstraction network $\phi_\varphi$ according to Eq. (5).
10:     Use the replay buffer $D$ and the TD3 [18] algorithm to update the policy and value network parameters $\theta$ and $\psi$.
11: **end for**

---

## C  Algorithm

The detailed algorithm workflow of AURO is in Alg. 1. The main differences between AURO and TD3 are: 1. The policy takes an additional latent variable $z_t$ as input . The latent variable is the output of the state abstraction module $\phi_\varphi$, which is trained with the loss specified in Eq. (5) 2. The exploration action $a_E$ is generated with Eq. (7) rather than by adding Gaussian noise to the original action $a_T$.

**Algorithmic Limitations and Failure Cases**  This paper focuses on the task of optimizing user retention, without considering the combination of retention signals with immediate user feedback. It will be interesting to investigate how to balance these two kinds of reward signals. One potential failure case of our algorithm is when users exhibit unusual or adversarial patterns that have never been witnessed by the algorithm. For example, users may suddenly get bored of the app and dislike every item that is recommended. Our state abstraction module may fail to adapt such behavior pattern and recommend appropriate items. Another case is stubborn users that do not click any novel item that they are unfamiliar with. Our exploration module may lead to inferior retention performance on such users, compared with greedy recommendation methods.

We would also like to mention that in our live recommendation platform, RL-based algorithms are only one step of the long recommendation pipeline. Items selected by AURO will not be directly presented to users before further ranking and filtering. The filtering mechanism ensure that no dangerous or illegal items will be presented to users. When applying similar RL-based recommendation algorithms to real-world, it is very important for future practitioners to be aware of post processing, such as item filtering, for safe and robust recommendation.

**Table 5: The hyperparameters for the AURO algorithm.**

|          | Hyperparameter | Value |
|----------|----------------|-------|
| Training | Optimizer | Adam |
|          | Learning rate | $3 \cdot 10^{-4}$ |
|          | Batch size $B$ | 256 |
| Network  | Number of transformer layers | 2 |
|          | Dimension of feedforward networks | 64 |
|          | Number of attention heads | 4 |
| RL       | Discount factor $\gamma$ | 0.9 |
|          | Reward weighing parameter $\lambda$ | -0.1 |
|          | Replay buffer size | $5 \times 10^4$ |
|          | Target smoothing coefficient | 0.005 |
|          | Target update interval | 1 |
|          | Action Space Dimension $k$ | 7 |
| AURO     | Number of clusters $n$ | 4 |
|          | $\beta$ in exploration | 30 |
|          | $\delta$ in exploration | 1 |
|          | Number of candidate actions | 6 |

**Table 6: Average time cost of action selection in one training step.**

|                        | Action Selection (s) | Total Time (s) |
|------------------------|----------------------|----------------|
| AURO                   | 0.259                | 1.738          |
| AURO (no exploration)  | 0.113                | 1.595          |
| Exploration Time Cost  | 129 %                | 8.966 %        |

## D  Additional Experiment Details

### D.1  Setup

According to KuaiSim [50], the network architecture of the retention simulator is similar to the policy network. We assume the immediate user response follows a Bernoulli distribution and the user return time in days follows a geometric distribution. The simulator is trained in a style of supervised learning and is updated by likelihood maximization on the training data. The hyperparameters for the AURO algorithm during training are specified in Tab. 5.

### D.2  Baselines

In the comparative experiments in the KuaiSim simulator, we consider the following baselines:

- CEM [13]: The Cross Entropy Method, which is commonly used as a surrogate for RL in recommendation tasks.
- DIN [53]: The Deep Interest Network that learns the representation of user interests from historical behaviors and performs click-through rate prediction. We keep the original input of DIN unchanged and use transformers, which are of the same structure with AURO, as the backbone network. This help us make fair comparisons.
- DDPG [26]: A value-based off-policy RL algorithm with a deterministic policy updated with deterministic policy gradients.

- TD3 [18]: A value-based off-policy RL algorithm that on top of DDPG, incorporates a pair of Q-networks to mitigate overestimation.
- SAC [19]: A value-based off-policy RL algorithm with a stochastic policy and the maximum-entropy RL objective.
- OAC [12]: An RL-based exploration algorithm that incorporates the idea of optimism under uncertainty, and obtains a separate optimistic action during the training phase.
- RND [5]: An RL-based exploration algorithm that encodes the state input to fit the output of a random network. States with larger encoding error will be assigned with higher intrinsic reward.
- ESCP [31]: The state-of-the-art environment sensitive contextual Meta-RL approach that explicitly identifies hidden parameters as additional policy inputs. During training, ESCP assumes access to the hidden parameters, which is unavailabe in live experiments. In practice, we use the level of user activeness as the proxy hidden parameter in ESCP.
- OSPIA [44]: An one-step policy improvement algorithm that biases the policy towards recommendations with higher long-term user engagement metrics.
- RLUR [6]: An RL-based recommendation algorithm especially designed for optimizing long-term user engagement.

### D.3  Computation Cost of the Exploration Module

AURO requires addtional steps in action selection, computing the gradient of the Q-function and sampling among candidate actions. But apart from action selection, RL training involves interacting with the environment and updating the policy with gradient decent. These two parts will take up more time than action selection. We conduct empirical studies and exhibit in Tab. 6 the average time cost of action selection in one training step. The total time cost of one training step is also shown for comparison. According to the results, although the exploration module lead to an addtional 129% of computation cost, it only costs less than 10% more total time. Also, during deployment the exploration module is not included, so it adds no more computation cost.

