# OpenReview forum: "AURO: Reinforcement Learning for Adaptive User Retention Optimization in Recommender Systems"
_ACM.org/TheWebConf/2025/Conference — WWW 2025 Oral_

### Official Review · Reviewer_aBMP · 2024-11-20

**Novelty:** 4
**Technical Quality:** 5

**Review:**

## Pros:

+ The paper introduces AURO, a novel approach addressing the challenge of non-stationarity in recommender systems through reinforcement learning.

+ The technical foundation of the paper is solid, utilizing a state abstraction module integrated within the policy network to handle environment shifts effectively.

+ AURO is shown to outperform existing methods significantly.

## Cons:

+ The success of AURO heavily depends on the accuracy of the state abstraction module. If the abstraction is not precise in representing the underlying environment state, the entire RL policy may become suboptimal. The paper does not explore failure cases or provide a fallback strategy when state abstraction fails.

+ Figure 2 illustrates the diverse behavioral patterns of users within the system. How does AURO perform across different user groups? It is recommended to conduct more in-depth analytical experiments.

**Questions:**

Please see the reviews.

**Reviewer Confidence:**

2: The reviewer is willing to defend the evaluation, but it is likely that the reviewer did not understand parts of the paper

**Scope:**

4: The work is relevant to the Web and to the track, and is of broad interest to the community

---

### Official Review · Reviewer_8oau · 2024-12-02

**Novelty:** 3
**Technical Quality:** 5

**Review:**

To address the issue of environmental non-stationarity in RL-based recommendation systems, this paper proposes the AURO method. AURO primarily introduces a state abstraction module within the policy network to actively identify RL policies, thereby reflecting changes in the environment and notifying the recommendation policy to make corresponding adjustments. AURO outperforms existing methods in optimizing long-term and non-fixed user retention, ensuring the stability of recommendation quality.

Pros:

1. The paper offers a valuable numerical analysis of the existing challenges, effectively demonstrating that manually extracting environmental information through predefined rules and integrating it into the policy network is increasingly difficult. This difficulty arises primarily from the inherent complexity and non-stationarity of the recommendation environment.
2. The experimental setup is relatively reasonable, with experimental analyses corresponding to the descriptions of key strategies. The experiments conducted effectively address the challenges proposed in the paper.
3. The paper's structure is well-organized, providing a coherent flow that facilitates understanding of the proposed methodology and its contributions.

Cons:

1. The discussion of "poor generalization ability" and "overly optimistic and unreliable exploratory actions" in other methods is not sufficiently detailed. The paper should elaborate on the underlying reasons for these shortcomings, supported by references to related studies or experimental evidence.
2. Figure 1 illustrates the general workflow of RL-based recommendation methods, but it does not clearly highlight the unique aspects of the proposed AURO method compared to existing approaches. Adding annotations or a comparative diagram to emphasize key distinctions would improve clarity.
3. The rationale for designing the abstraction net, as described in the third paragraph of the proposed method section, lacks a clear and strong logical connection. It is necessary to explain why the abstraction net is essential, particularly in addressing the sensitivity of the policy to environmental changes.
4. While the abstract mentions two challenges, the methodology section addresses only one. This discrepancy needs to be explained, and the paper should clarify whether "non-stationarity environment" and "complex environment changes" refer to the same concept or represent distinct challenges. If they are different, the discussion should explicitly distinguish between them.
5. Validating the proposed method on a single dataset limits the generalizability of the findings. Including one or two additional datasets, especially those reflecting diverse user behavior patterns or environmental dynamics, would better demonstrate the robustness and effectiveness of AURO.

**Questions:**

1. The paper highlights issues such as "poor generalization ability" and "overly optimistic and propose unreliable exploratory actions" in other methods. However, the underlying reasons for these shortcomings are not sufficiently elaborated.
2. Figure 1 presents the general workflow for RL-based recommendation methods; however, the unique aspects of the proposed method compared to existing approaches are not explicitly highlighted.
3. In the third paragraph describing the proposed method, it is stated that the policy's sensitivity to changes necessitates the design of an abstraction net for active identification. However, the reasoning behind this decision lacks a clear and strong logical connection.
4. The abstract mentions two challenges, but the method section addresses only one. The reason for this discrepancy is unclear and should be explained. Additionally, the terms "non-stationarity environment" and "complex environment changes" are used, but it is not clear whether they refer to the same concept or represent distinct challenges.
5. Validating the method with only one dataset may limit the generalizability of the findings. It is recommended to include one or two additional datasets to better demonstrate the robustness and effectiveness of the proposed method.

**Reviewer Confidence:**

2: The reviewer is willing to defend the evaluation, but it is likely that the reviewer did not understand parts of the paper

**Scope:**

4: The work is relevant to the Web and to the track, and is of broad interest to the community

---

### Official Review · Reviewer_quX7 · 2024-12-02

**Novelty:** 3
**Technical Quality:** 5

**Review:**

Summary:

This paper introduces AURO (Adaptive User Retention Optimization), a reinforcement learning framework designed to optimize user retention in recommender systems operating in non-stationary environments. AURO uses a state abstraction module to addresses the challenges of non-stationary user behaviors. Additonally, it employs a guarded exploration mechanism, which combines optimism under uncertainty with rejection sampling to encourage exploration while maintaining a stable recommendation quality. The approach is validated empirically using a user retention simulator, the MovieLens dataset, and live experiments on a real-world short-video platform.

Pros:

1. The focus on optimizing user retention and addressing non-stationarity in user behavior for RL applications is well-motivated.
2. The proposed method is evaluated across multiple settings: a simulator, a real-world dataset, and large-scale live experiments, demonstrating its practical effectiveness.
3. Comprehensive ablation studies and analyses highlight the contributions of each module.
4. The source code is provided.

Cons:

1. While addressing non-stationarity in RL is a well-known challenge, the fundamental idea of detecting environmental changes and adapting the model is not new. The method proposed in this paper appears to follow this general framework, and the novelty compared to existing techniques is not explicitly articulated. The authors should clarify the novelty of their method and its differences from existing approaches.
2. The paper frequently uses abstract and convoluted phrases rather than simple and concrete language, which affects the readability. For example, terms like "value-based state abstraction", "cost-sensitive recommendation", "sparse retention signal", "well-aligned state abstraction network", "abstraction-performance alignment", "performance-based rejection sampling". These terms may be familiar to experts in the field or assumed to be self-explanatory. However, they can confuse readers unfamiliar with the topic and require readers to guess their meaning when first encountered. I recommend that the authors use simpler and more specific language whenever possible to improve clarity.
3. Some minor issues:\
    Line 19: an state -> a state\
    Line 190: The authors may want to use `\langle` and `\rangle` instead of `<` and `>`.

**Questions:**

1. Could the authors elaborate on the technical novelty of their work? Specifically, how do the state abstraction module and the guarded exploration mechanism differ from existing techniques for handling non-stationary environments?
2. In Section 2, the retention reward is proportional to user return time. This seems counterintuitive, as a larger user return time implies lower user activity. Could the authors clarify the reasoning behind this formulation?

**Reviewer Confidence:**

2: The reviewer is willing to defend the evaluation, but it is likely that the reviewer did not understand parts of the paper

**Scope:**

4: The work is relevant to the Web and to the track, and is of broad interest to the community

---

### Official Review · Reviewer_pzN4 · 2024-12-03

**Novelty:** 7
**Technical Quality:** 7

**Review:**

This paper introduces AURO (Adaptive User Retention Optimization), a novel reinforcement learning approach for recommender systems that specifically addresses the challenge of non-stationary environments where user behavior patterns, such as interaction rates and retention propensities, constantly evolve over time.
The motivation of the work is that existing RL algorithms for recommendations often struggle to adapt to these dynamic environments, resulting in performance degradation as user behavior shifts. AURO is proposed to overcome this limitation by enabling recommendation policies to dynamically adapt to these changes.
The idea is that AURO introduces a state abstraction module within the policy network. This module is trained using a unique value-based loss function that aligns its output with the estimated performance of the current policy. By capturing information about the changing environment, the state abstraction module signals the policy to adapt accordingly. In addition, to further enhance adaptation, AURO incorporates a guarded exploration strategy. This strategy employs optimism under uncertainty to encourage exploration but combines it with performance-based rejection sampling to ensure that only potentially rewarding exploratory actions are taken, promoting stability in the online training process.
In the paper, AURO's performance is thoroughly evaluated using three distinct approaches:
1.User Retention Simulator: The KuaiSim retention simulator, designed to mimic user retention on short-video platforms, was used to test AURO in a controlled environment. The simulator incorporates dynamic probabilities for user session termination and return times, creating a non-stationary environment. Here, AURO was compared against various baseline algorithms, including traditional recommendation methods, state-of-the-art value-based RL algorithms, exploration-focused RL algorithms , context encoder-based algorithms, and RL algorithms specifically designed for user retention optimization. Metrics on which the approaches were compared agains were Average return days, next-day return probability, and retention reward (calculated based on return time). It was shown that AURO consistently outperformed all baselines, demonstrating its effectiveness in optimizing user retention within a dynamic environment.
2. MovieLens Dataset: The well-known MovieLens-1M dataset, containing movie ratings, was used to evaluate AURO on a different recommendation task. Retention rewards were derived from movie ratings based on the assumption that higher ratings correlate with longer return times. It was shown that AURO, with the modifications for offline training, achieved the highest retention reward compared to baseline algorithms.
3. Live A/B Test: AURO was deployed in a live A/B test on a large-scale, real-world short-video recommendation platform. Millions of users were randomly assigned to different groups exposed to AURO, TD3, ESCP, and the baseline algorithm RLUR. Metrics used were 7-day retention rate, average application dwell time, click-through rate, like rate, comment rate, and unlike rate. The results indicate that AURO showed significant improvements over all baseline algorithms across all evaluation metrics. Notably, it was the only algorithm to improve dwell time and showed substantial gains in comment rate and a reduction in the unlike rate, suggesting both increased user engagement and satisfaction.

Strengths of the Paper
1. Innovative Idea: The introduction of a value-based state abstraction module to address non-stationary environments in recommendation systems is a novel and promising approach.
2. Technical Soundness: The paper provides a clear and detailed explanation of the AURO algorithm, its theoretical foundations, and its implementation.
3. Thorough Evaluation: The use of a simulator, an established dataset, and a real-world A/B test provides strong evidence for AURO's effectiveness and generalizability.

A Note on the MovieLens Evaluation
While not necessarily a weakness, the evaluation of AURO on the MovieLens dataset, which only contains rating data, might be considered an unnecessary addition given the comprehensive live evaluation. The assumption that ratings directly correlate with return times, while plausible, might not fully capture the nuances of real-world user behavior. The live A/B test results arguably provide more compelling and practical insights into AURO's performance in a dynamic, non-stationary environment.

**Questions:**

Given my previous statement on the evaluation using the MovieLens dataset (and the whole approach), even though others were doing this as well, what are the benefits of this evaluation - what new insights did this evaluation provided to you?

**Reviewer Confidence:**

2: The reviewer is willing to defend the evaluation, but it is likely that the reviewer did not understand parts of the paper

**Scope:**

4: The work is relevant to the Web and to the track, and is of broad interest to the community

---

### Official Review · Reviewer_tE89 · 2024-12-05

**Novelty:** 3
**Technical Quality:** 4

**Review:**

This paper presents the Adaptive User Retention Optimization (AURO) algorithm, addressing the challenges of non-stationary environments in recommendation systems. By integrating a state abstraction module and a value-based contrastive loss function, AURO effectively identifies changes in user behavior and adapts policies for improved user retention.

Pros:
The introduction of the state abstraction module and value-based loss function provides a fresh perspective on handling non-stationary environments in recommendation systems.

The writing is generally clear and concise, with a logical flow that guides the reader through the problem statement and methodology.

The experiments conducted on real-world datasets demonstrate the effectiveness of AURO compared to baseline algorithms, showcasing its practical applicability.

Cons:
The proposed method may require sophisticated implementation and tuning, which could be a barrier for practitioners looking to adopt the approach.

The paper may be of somewhat narrow application and challenging for readers without a strong background in RL, potentially limiting practicality.

While the results are promising, the performance of AURO may vary across different domains or datasets, and further validation in diverse contexts would strengthen the claims.

**Questions:**

Q1: How does AURO incorporate user feedback over time? Are there other mechanisms in place to adjust recommendations based on long-term user preferences?

Q2:Can you elaborate on how AURO balances exploration and exploitation, particularly in scenarios where user behavior is highly unpredictable? What strategies do you recommend for optimizing this balance?

Q3:What specific performance metrics did you prioritize in your evaluations, and how do these metrics align with real-world user engagement goals?

**Reviewer Confidence:**

2: The reviewer is willing to defend the evaluation, but it is likely that the reviewer did not understand parts of the paper

**Scope:**

3: The work is somewhat relevant to the Web and to the track, and is of narrow interest to a sub-community